# Crosslinked Pore-Filling Anion Exchange Membrane Using the Cylindrical Centrifugal Force for Anion Exchange Membrane Fuel Cell System

**DOI:** 10.3390/polym12112758

**Published:** 2020-11-23

**Authors:** Tae Yang Son, Tae-Hyun Kim, Sang Yong Nam

**Affiliations:** 1Department of Materials Engineering and Convergence Technology, Engineering Research Institute, Gyeongsang National University, Jinju 52828, Korea; kr6620@naver.com; 2Oragnic Material Synthesis Laboratory, Department of Chemistry, Incheon National University, Incheon 22012, Korea; tkim@inu.ac.kr

**Keywords:** anion exchange membrane fuel cell, anion exchange pore-filled membrane, poly(phenylene oxide), quaternary ammonium, centrifugal force pore-filling method

## Abstract

In this study, novel crosslinked pore-filling membranes were fabricated by using a centrifugal force from the cylindrical centrifugal machine. For preparing these crosslinked pore-filling membranes, the poly(phenylene oxide) containing long side chains to improve the water management (hydrophilic), porous polyethylene support (hydrophobic) and crosslinker based on the diamine were used. The resulting membranes showed a uniform thickness, flexible and transparent because it is well filled. Among them, PF-XAc-PPO70_25 showed good mechanical properties (56.1 MPa of tensile strength and 781.0 MPa of Young’s modulus) and dimensional stability due to the support. In addition, it has a high hydroxide conductivity (87.1 mS/cm at 80 °C) and low area specific resistance (0.040 Ω·cm^2^), at the same time showing stable alkaline stability. These data outperformed the commercial FAA-3-50 membrane sold by Fumatech in Germany. Based on the optimized properties, membrane electrode assembly using XAc-PPO70_25 revealed excellent cell performance (maximum power density: 239 mW/cm^2^ at 0.49 V) than those of commercial FAA-3-50 Fumatech anion exchange membrane (maximum power density: 212 mW/cm^2^ at 0.54 V) under the operating condition of 60 °C and 100% RH as well. It was expected that PF-XAc-PPO70_25 could be an excellent candidate based on the results superior to those of commercial membranes in these essential characteristics of fuel cells.

## 1. Introduction

As climate change accelerates, the development of green energies has attracted significant attention [1,2]. Among them, fuel cells can convert chemical energy into electrical energy and have the advantages of high efficiency and eco-friendly [3,4,5,6]. There are several types of fuel cell systems, according to the kinds of electrolyte. Especially, a polymer electrolyte membrane fuel cell (PEMFC) is a fuel cell system using the proton exchange membrane as an electrolyte. However, a proton conductivity is also rapidly decreasing in a high temperature and low humidity environment. The high cost of platinum catalysts and electrolyte membranes is also still a problem. Compared to the PEMFC, an anion exchange membrane fuel cell (AEMFC) has operating advantages such as lower activation energy for the oxygen reduction reaction (ORR) in an alkaline medium and inexpensive cost of non-platinum catalysts and membranes [7]. Despite these advantages, the practical application of AEMFC faces challenges. The hydroxide conductivity of anion exchange membranes is low due to the its lower mobility. For the reason, the anion exchange membrane requires a higher ion exchange capacity (IEC) than the proton exchange membrane. The high conductivity due to the high IEC and low dimensional stability due to the swelling always exist in a trade-off relationship. In other words, the low conductivity and poor mechanical and chemical stability of anion exchange membranes are still the weakness [8,9,10]. So, controlling the swelling and stability is critical to improve the conductivity and stability directly related to the cell performance [11]. Nevertheless, a number of recent studies have been conducted, and remarkable performance improvements have been made [12,13,14,15,16,17,18,19,20]. There is a need to continue further research in order to achieve comparable performance to Nafion membrane that is used in proton exchange membrane fuel cell.

Therefore, improvement of conductivity and stability through various techniques is important in the AEMFC [4,12,13,14]. First, many polymers are being studied for use in anion exchange polymers. Examples include poly(arylene ether sulfone), poly(ether ether ketone), poly(styrene-ethylene-butylene-styrene), poly(styrene-butadiene-styrene), polybenzimidazole, poly(phenylene), poly(phenylene oxide) and polynorbornene [21,22,23,24,25,26,27,28,29,30,31,32]. The anion exchange polymers with long side chain came up as a remarkable revolution to improve the water uptake because it can trap water between the polymer chains mentioned above [33]. Second, the pore-filled membrane is good approach that can improve the mechanical stability by using the physical property of the porous support [4]. Third, recently, for the chemical stability of the anion exchange membrane, the hydrophobic and hydrophilic parts are phase separated. When a pore-filled membrane is manufactured using a hydrophobic porous support, chemical stability is improved because phase separation can be artificially induced [34]. The pore-filled membrane has these advantages, but it should be formed the thinner and more uniformly for high conductivity and excellent cell performance in the AEMFC. So, in order to fill the hydrophilic polymer into the hydrophobic porous support, a pretreatment process or an additional process is required.

The present work focused on the improvement of water management, conductivity, mechanical and chemical stability of anion exchange membranes based on the pore-filled membrane using the poly(phenylene oxide) with long side chain and quaternary ammonium group through Friedel crafts acylation reaction, and polyethylene (PE) support [35].

Even with the above solution, chemical stability problems due to (i) Hofmann elimination (E2), and (ii) nucleophilic substitution (S_N_2) may remain [36,37]. So, the crosslinking is a good option to solve the chemical stability using the crosslinker based on the diamine.

In addition, if the hydrophilic polymer is filled in the hydrophobic support using the physical force using centrifugal force, it is easy for anyone to fabricate it without pretreatment or additional processes. The final membranes were designated as the crosslinked pore-filling membranes, and then, the properties including conductivity, stability, and cell performance were discussed here.

## 2. Materials and Methods

### 2.1. Materials

Poly(2,6-dimethyl-1,4-phenylene oxide) (PPO) was purchased from Asahi Kasei Corp (Tokyo, Japan). 6-bromohexanoyl chloride, aluminum chloride, 1,2-dichloroethane anhydrous, trimethyl ammonium solution (45 wt% in DI water) and *N*,*N*,*N’N’*-tetramethyl-1,6-hexanediamine (TMHDA) were obtained from Sigma Aldrich Korea (Yongin, Korea). Methyl alcohol (MeOH), *N*-methyl-2-pyrrolidone (NMP) and potassium hydroxide (KOH) were purchased from Daejung Chemical (Siheung, Korea). The polyethylene (PE) support was provided from W-SCOPE KOREA (Ochang, Korea).

### 2.2. Synthesis of Acylated Poly(Phenylene Oxide) (Ac-PPO)

The acylated poly(phenylene oxide) (Ac-PPO) was synthesized in a 3-neck round bottom flask with dropping funnel. The PPO solution was prepared by dissolving poly(2,6-dimethyl-1,4-phenylene oxide) (PPO, 5.00 g, 41.61 mmol) in 1,2-dichloroethane anhydrous solvent. At the same time, the reagent solution for Friedel-Crafts acylation reaction was prepared using 6-bromohexanoyl chloride and fecalaluminum chloride. More details, fecalaluminum chloride (AlCl_3_, 4.27 g, 32.04 mmol) was dispersed in the 1,2-dichloroethane anhydrous solvent with nitrogen gas at the one neck round bottom flask, and then, 6-bromohexanoyl chloride (6.84 g, 32.04 mmol) was dropped slowly in the dispersed solution using a dropping funnel. The reagent solution was obtained after 30 min. Then, this solution was inserted to the PPO solution using dropping funnel at the ice bath. After fully dropping the reagent solution, the final solution was maintained at the room temperature for 6 h, and the solution was precipitated using methyl alcohol, the Ac-PPO obtained was washed several times in the same non-solvent. Finally, the product was dried in a vacuum oven at 60 °C for 24 h, resulting in Ac-PPO70 at a yield of 94.71%.

### 2.3. Preparation of Crosslinked Pore-Filling Anion Exchange Membrane

For preparing the pore filled anion exchange membrane, Ac-PPO was dissolved in THF to a concentration of 15 wt%. The TMHDA crosslinking agent was inserted before impregnation by controlling the degree of crosslinking to prepare a crosslinked polymer solution. The porous PE support was extended in a cylindrical centrifugal machine, after which the prepared polymer solution poured into the same machine. Then, the polymer solution penetrated into the pores of the PE support by centrifugal force. The solvent of polymer solution was evaporated for 12 h with IR ramp at the same time as the machine was driven. As the solvent is volatilized, the polymer is filled to a uniform thickness in the pores of the support. After fully dried, the pore-filled membrane was soaked in the 45% trimethylamine solution (TMA solution) to modify into conducting group for 24 h. This is because the modifying reaction is a heterogeneous condition, it needs enough replacing time. The pore-filled membrane containing conducting group was subjected to alkalization in 1M KOH solution for 24 h at room temperature. The OH– exchanged membranes were washed several times with deionized water to remove any residual KOH and stored in DI water until the characterization process. The pore-filled membrane using PE substrate is named as PF-XAc-PPOX_Y membrane, where, X represents the degree of substitution (DS) of quaternary ammonium group and Y represent the degree of crosslinking, respectively. For example, PF-XAc-PPO70_25 is prepared as a pore-filling membrane using a PE support in a 25% crosslinked solution of an Ac-PPO polymer having a degree of substitution of 70%.

### 2.4. Experimental Technique

#### 2.4.1. ^1^H-NMR and FE-SEM

The ^1^H-NMR spectra of Ac-PPO were obtained on DRX300 (300 MHz) (Bruker, Bilerica, MA, USA). Chloroform-d was used as the solvent.

The morphology of the crosslinked pore-filling membranes was observed by means of a FE-SEM (Field-Emission Scanning Electron Microscope, Philips XL30S FEG, Amsterdam, The Netherlands). The observations of the pore-filled membranes consisted of surface and cross-section assessments.

#### 2.4.2. Ion Exchange Capacity, Water Uptake, Swelling Ratio and Hydration Number

The back-titration method was used for the anion exchange membrane. The pore-filled membrane was immerged in a 0.01M HCl solution for 24 h, and the IEC value was measured by titration using a 0.01M NaOH solution. After finishing the titration process, the pore-filled membrane was removed and dried in a vacuum oven at 60 °C. The weight of the dried pore-filled membrane was then measured. Finally, the ion exchange capacity was calculated using the following equation:(1)$$IEC\;\left(meq/g\right)=\;\frac{\left[\left(V_{HCl}\times M_{HCl}\right)-\;\left(V_{NaOH}\times M_{NaOH}\right)\right]}{W_{dry}}$$
where *M_HCl_* (M) and *V_HCl_* (mL) are correspondingly the concentration and volume of the initial HCl solution. *M_NaOH_* (M) and *V_NaOH_* (mL) are the concentration and volume of standard NaOH solution used for titration, respectively. *W_dry_* (g) is the weight of the dry crosslinked pore-filling membrane.

The water uptake was calculated according to the weight change of the crosslinked pore-filling membrane. To determine the water uptake value, the crosslinked pore-filling membrane was cut into an appropriate size, and then weighed (*W_dry_*). These were then immersed in DI water for 24 h. Excess water on the sample surface was wiped away with tissue paper and weighed (*W_wet_*). The water uptake was calculated using the following equation:(2)$$Water\;uptake\;\left(\%\right)=\;\frac{\left(W_{wet}-W_{dry}\right)}{W_{dry}}\times 100$$

The hydration number of water molecules per ionic group (*λ*) was determined using the following equation:(3)$$Hydration\;number\;\left(\lambda \right)=\;\frac{\left(W_{wet}-W_{dry}\right)/W_{dry}}{18\;\times IEC}$$

Swelling ratio (%) of pore-filled membranes were measured by immersing the rectangle shape membranes into deionized water, and the changes of size were calculated using the following equation:(4)$$Swelling\;ratio\;\left(\%\right)=\;\frac{\left(L_{wet}-L_{dry}\right)}{L_{dry}}\;\times 100$$
where *L_wet_* and *L_dry_* represent the length of the wet and dry crosslinked pore-filling membrane, respectively.

#### 2.4.3. Mechanical Properties: Tensile Strength, Elongation at Break and Young’s Modulus

The mechanical properties of the crosslinked pore-filling membrane were measured by using a universal testing machine (UTM, LR10K at LLOYD, Berwyn, PA, USA) according to the ASTM D638 5. To confirm the mechanical properties, the tension speed was 10 mm/min.

#### 2.4.4. Hydroxide Conductivity and Area Specific Resistance

The hydroxide conductivity of the pore-filled membrane is an important factor. The hydroxide conductivity was determined using the measured value of the membrane resistance. The hydroxide conductivity tests of the pore-filled membranes were carried out at from 25 to 80 °C and at a relative humidity of 100% by the impedance method using electrochemical spectroscopy (SP-300, Bio Logic Science Instrument, Seyssinet-Pariset, France). Finally, the hydroxide conductivity was calculated using the following equation:(5)σ= L/R ×A

σ is the ion conductivity, *R* is the electrical resistance, and *L* and *A* are respectively the thickness and area of the membrane. The ionic area specific resistance (*ASR*) was calculated using the following equation:(6)ASR= T/σ

*T* is the thickness of the crosslinked pore-filling membranes and σ is the hydroxide conductivity at 80 °C.

#### 2.4.5. Alkaline Stability

The alkaline stability of the crosslinked pore-filling membranes was investigated by soaking the OH− form membranes into 1M KOH solution in 60 °C for 800 h to evaluate the changes in ionic conductivity. Before measurements, each membrane was washed with deionized water several times to remove the free KOH inside the membrane. The ionic conductivity of each membrane was measured in high purity deionized water at room temp.

#### 2.4.6. Single Cell Performance

Single cell performances using the membranes were analyzed at 60 °C under 100% RH and H_2_ and O_2_ feed flow rates of 300 mL/min and 300 mL/min respectively. The catalyst slurry was prepared using a mixture of Pt/C catalyst (46.2 wt%, Tanaka, Japan), 2-propanol and ionomer solution (Sustainion XB-7, 5 wt% in ethanol, Dioxide Materials, Boca Raton, FL, USA) by ultra-sonicating method. The CCMs (Catalyst Coated Membranes) were prepared by spraying the slurry on the membrane (1.0 mg Pt/cm2) with an air spray gun. The MEA (Membrane Electrode Assembly) was sandwiched between gas diffusion layer (GDL, Sigracet 39BC) and gasket (Teflon, CNL energy) and the effective area was 9 cm^2^.

## 3. Results and Discussions

### 3.1. Structure Analysis

Figure 1 shows the preparation of the acylated PPO (Ac-PPO). The Ac-PPO was synthesized by Friedel-craft acylation reaction. The structure of the synthetic polymer was confirmed by ^1^H-NMR spectroscopy.

Figure 2 shows the ^1^H-NMR spectrum of the Ac-PPO polymer. In the Figure 2, the peak at 6.45 ppm was assigned to the aromatic proton peak of the PPO polymer (2H, Ar-H). Following Friedel-craft acylation, the proton peak of the acylated aromatic proton showed at the 6.06 ppm (1H, Ar-H). Furthermore, the peaks at 2.93 and 3.36 ppm were assigned to the additional protons adjacent to the carbonyl group (2H,–CH_2_–) and the bromine (2H,–CH_2_–). In addition, the degree of substitution was determined from the integral area values ratio of the peak at 6.45 and 6.06 ppm [35].

To obtain the chemical stability of the pore-filled membrane, crosslinking was performed using a halogen element and a diamine crosslinking agent, and the degree of crosslinking was adjusted through the amount of the crosslinking agent. Figure 3 is the schematic image of crosslinking.

### 3.2. Fabrication of Crosslinked Pore-Filling Membrane and Morphology

In Figure 4, photographs and FE-SEM images of the prepared crosslinked pore-filling membrane and PE support were compared. First, it was confirmed that a transparent cross-linked pore-filling membrane was prepared by impregnating a polymer solution on an opaque support through the cylindrical centrifugal machine (Figure 5). Continue, the FE-SEM images of the PE support before and after pore filling are shown in Figure 4. In particular, it was confirmed that the pores in the porous support disappeared in PF-XAc-PPO70_25 from the surface and cross-sectional photographs of the FE-SEM. As a result, it was possible to ensure that the ion exchange polymer was well filled (Appendix A). In addition, the average thickness of the prepared crosslinked pore-filling membrane was 30 ± 2 μm.

In the anion exchange membrane fuel cell, since local hydration of the too thin electrolyte membrane less occurs, the resistance increases, resulting in performance degradation [15]. For that reason, it is judged that the thickness of the prepared crosslinked filler film was appropriately increased in a 20 μm support and fabricated.

Figure 6 shows the photographs of the crosslinked pore-filling membranes with and without a cylindrical centrifugal machine.

As can be seen from Figure 6b, there is a part where the hydrophilic polymer is difficult to impregnate the hydrophobic support. However, the ion exchange polymer solution was well filled by using a centrifugal force in the porous support. As a result, the crosslinked pore-filling membrane became transparent in the Figure 6c.

### 3.3. Mechanical Properties

To improve the durability of the membrane electrode assembly (MEA), the mechanical strength of the crosslinked pore-filling membrane is an important factor. Table 1 shows the tensile strength, elongation at break and Young’s modulus.

In the case of XAc-PPO70_100, its physical properties were so weak that it could be crushed when touched. However, the crosslinked pore-filling membranes showed excellent mechanical properties. In particular, PF-XAc-PPO70_25 showed a tensile strength of 56.1 MPa, an elongation at break of 32.7%, and a Young’s modulus of 781.0 MPa. It was confirmed that the elongation gradually decreased as the degree of crosslinking increased as reported by other studies [38,39,40,41]. Based on these results, it was confirmed that the crosslinked pore-filling membrane was a good option [4].

### 3.4. IEC, WU, SR and λ value

Table 2 shows the ion exchange capacities (IEC) of the XAc-PPO70_100 and the prepared crosslinked pore-filling membranes.

In the case of XAc-PPO70_100, IEC was calculated as 2.96 meq/g based on the ^1^H-NMR spectra and was similar to the experimental value. The IEC values of all the PF-XAc-PPO70_Y showed lower than the XAc-PPO70_100 relatively. The lower IEC values observed for the crosslinked pore-filling membranes were ascribed to the porous PE support. In the case of IEC, since the unit is “meq/g”, it is affected by weight. However, there is no ion conducting group per unit weight in the case of the porous support, so the IEC of crosslinked pore-filling membranes with the support are relatively low [4]. As can be seen in the Table 2, nevertheless, no significant difference in the IEC was noted when varying the degree of crosslinking. The reason is believed to be that the conductive group is almost preserved by using a diamine-based crosslinking agent.

The water uptake and swelling ratio of the crosslinked pore-filling membranes in hydroxide form at 25 and 80 °C are depicted in Table 2. As expected, the water uptake and swelling percentage of the crosslinked pore-filling membranes increased with temperature. Moreover, the water uptake and swelling ratio are in inverse proportion to the degree of crosslinking in the Table 2. It is judged that these results were due to the improvement of water mobility and holding between polymer chains at the high temperature and low degree of crosslinking [38,42].

In addition, water management of the electrolyte membrane is essential for fuel cell performance and durability of the entire system in AEMFC [5]. At this time, since the hydration number is a factor that determines how many water molecules are collected around the ion conducting group, it is possible to judge how well water management of the pore-filled membrane is being performed. In this work, the high hydration number (λ) can be confirmed at the high temperature and low degree of crosslinking. The reason is that water molecules are collected around the hydrophilic anion conducting group at the end of the long side chain, and the water molecules are trapped due to the long side chain. In addition, as the degree of crosslinking decreases, it is judged that more water molecules are collected due to the overall chain flexibility. It can be seen that there are many water molecules around the ion conducting group, which can lead to good water management [5,42,43,44].

### 3.5. Hydroxide Conductivity & ASR

The hydroxide conductivity is an important factor on which the fuel cell performance and efficiency depend. The hydroxide conductivities of the crosslinked pore-filling membranes with different degree of crosslinking were measured at 25, 40, 60 and 80 °C. As shown in Figure 7A, the conductivities tended to be proportional to temperature. In addition, as the degree of crosslinking was lowered, it was found that the conductivity of the membrane was improved due to the flexibility of the polymer and the ease of water molecule mobility [33].

Moreover, the conductivity under humidified conditions did not differ significantly when compared to the conductivity in water (Table 3). Continuously, Figure 7B demonstrates the Arrhenius plot for the estimation of activation energy (Ea) required for hydroxide conduction through the anion exchange membrane. Activation energy was calculated using the slop of the regression line of ln σ vs 1000/T (K-1) as per the Arrhenius equation published elsewhere [45,46].

The slop values are 7.99 KJ/mol, 9.55 KJ/mol and 9.85 KJ/mol respectively for PF-XAc-PPO70_100, PF-XAc-PPO70_50 and PF-XAc-PPO70_25. As can be seen from the two graphs, low activation energy leads to high ionic conductivity. Based on the above results, the E_a_ values of crosslinked pore-filling membranes are in the range of 9.85 to 7.99 KJ/mol. These values are comparatively better than those values of previously reported anion exchange membranes (12.8 KJ/mol for Nafion-117 of Dupont [47], 27.4 KJ/mol for FAA-3-50 of Fumatech [48] and 13.0 KJ/mol for A201 of Tokuyama [49]).

Recently, the area specific resistance (ASR) is considered as important data for fuel cells as well as hydroxide conductivity. As a clear example of this, the US Department of Energy is targeting an ASR of 0.04 Ω·cm^2^ or less. The ionic ASR values for each crosslinked pore-filling membrane can be found in Table 3. As can be seen in the Table 3, the ASR of the PF-XAc-PPO70_25 is 0.040 Ω·cm^2^ [32].

### 3.6. Alkaline Stability

The chemical stability as known as alkaline stability of anion exchange membrane under alkaline medium conditions is related to the performance stability of AEMFC. In an alkaline atmosphere, the polymer main chain or conducting group may be degraded due to the S_N_2 reaction and the Hofmann elimination reaction. An alkaline stability was tested with the crosslinked pore-filling membranes in 1M KOH solution at 60 °C for 800 h.

As can be seen in the Figure 8, the decrease in conductivity after 800 h of the PF-XAc-PPO70_25 was small compared to other samples. In detail, it decreased by 24% compared to the initial conductivity value. As observed in Figure 8, PF-XAc-PPO70_25 exhibited about 76% of initial conductivity value after exposed in 1M KOH at 60 °C for 800 h could be due to the steric hindrance offered by the crosslinking structure which can reduce the access of nucleophilic attack to the functional groups by the excess amount of OH− and protect the backbone [50]. Moreover, the presence of water also has important role in hydroxide ion attack on membranes. PF-XAc-PPO70_25 is found to have a better alkaline stability than PF-XAc-PPO70_50 and PF-XAc-PPO70_100. The higher water uptake and improved solvation environment for OH– in PF-XAc-PPO70_25 would retard the degradation of cation and result in higher alkaline stability [51].

This is believed to be due to the fact that water management is possible due to a relatively small number of crosslinking, and thus the attack on the main chain is less [9].

### 3.7. Single Cell Performance

Finally, a single cell test was conducted to check its applicability in the AEMFC using the PF-XAc-PPO70_25 with the overall best properties in H_2_/O_2_ feed gas under 100% RH and 60 °C conditions. As can be seen in Figure 9, the MEA using PF-XAc-PPO70_25 showed an open circuit voltage (OCV) of 0.96 V and a peak power density of 239 mW/cm^2^ at ad current density of 549 mA/cm^2^. These results were much higher than those obtained from the commercial Fumatech FAA-3 membrane (392 mA/cm^2^ of current density and 212 mW/cm^2^ of maximum power density). (In the case of the Pt catalyst loading of 0.4 mg/cm^2^, a current density of 266 mA/cm^2^ and a peak power density of 114 mW/cm^2^ were shown [4].)

These good cell performance of our PF-XAc-PPO70_25 is attributed to the improved water management, conductivity, mechanical and chemical stability. Therefore, the PF-XAc-PPO70_25 was suggested an efficient anion exchange membrane candidate as an electrolyte membrane for the AEMFC application.

## 4. Conclusions

Crosslinked pore-filling membranes were developed by a cylindrical centrifugal machine. To improve the water management, long side chains were introduced in poly(phenylene oxide) through the Friedel-craft acylation reaction. At the same time, crosslinking was performed using a diamine crosslinker, and the porous PE support was well filled using a centrifugal force. All of the crosslinked pore-filling membranes exhibited good water uptake, swelling ratio, mechanical strength, hydroxide conductivity and chemical stability. Specifically, PF-XAc-PPO70_25 showed a hydroxide conductivity of 87 mS/cm at 80 °C in the DI water and 79.3 mS/cm at 80 °C in the humidity condition. The single cell performance of a membrane electrode assembly (MEA) employing the PF-XAc-PPO70_25 was found to be comparable to that employing a commercial membrane (FAA-3-50) due to good water management and physicochemical stability. Therefore, based on the above data, we believe that our crosslinked pore-filling membranes are promising candidates for anion exchange membrane fuel cell (AEMFC).

## Figures and Tables

**Figure 1 polymers-12-02758-f001:**
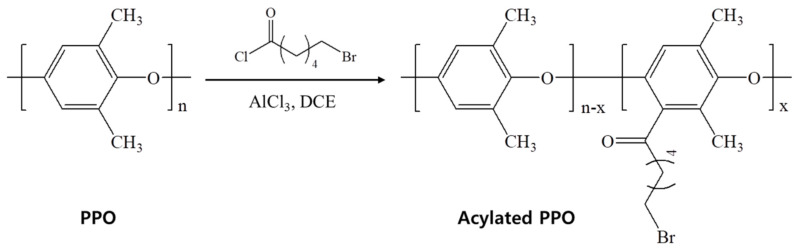
Scheme of the acylated poly(phenylene oxide) (Ac-PPO).

**Figure 2 polymers-12-02758-f002:**
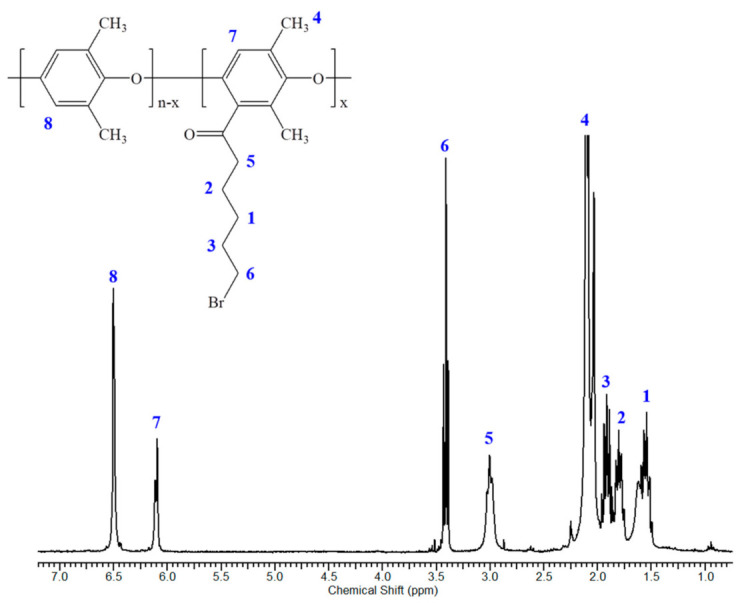
^1^H-NMR spectrum of the acylated poly(phenylene oxide) (Ac-PPO): Ac-PPO70.

**Figure 3 polymers-12-02758-f003:**
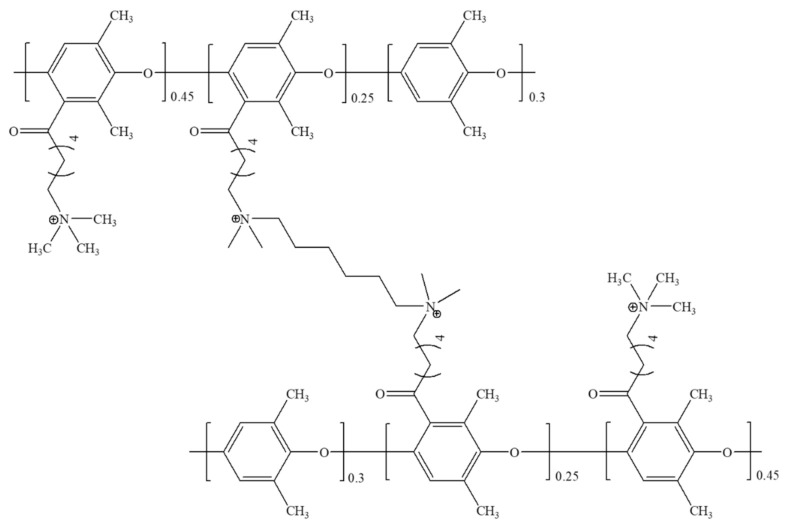
Scheme of the acylated poly(phenylene oxide) with the crosslinking (XAc-PPO).

**Figure 4 polymers-12-02758-f004:**
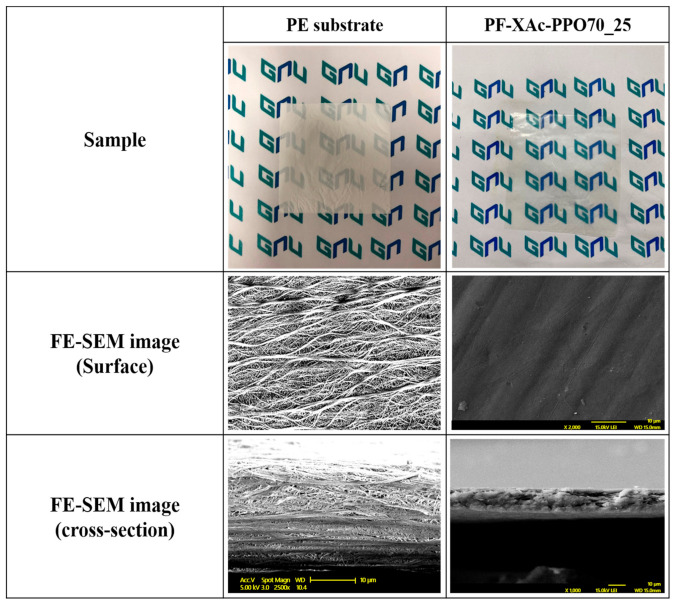
FE-SEM images of the surface and cross–section of the porous PE support and crosslinked pore-filling membrane (PF-XAc-PPO70_25).

**Figure 5 polymers-12-02758-f005:**
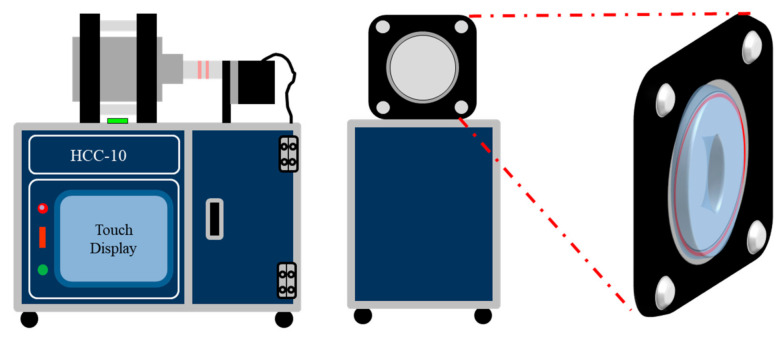
Apparatus for preparing pore-filled membrane: cylindrical centrifugal machine.

**Figure 6 polymers-12-02758-f006:**
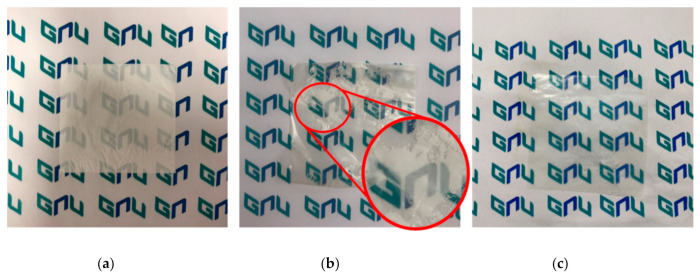
The photo comparison: (**a**) porous polyethylene support, (**b**) crosslinked pore-filling membrane without a cylindrical centrifugal machine, and (**c**) crosslinked pore-filling membrane with the cylindrical centrifugal machine.

**Figure 7 polymers-12-02758-f007:**
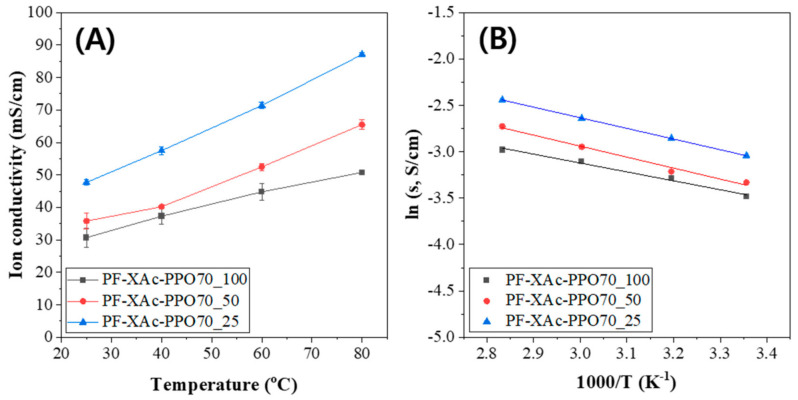
(**A**) Hydroxide conductivity and (**B**) Arrhenius plot of crosslinked pore-filling membranes in the temperature rising process.

**Figure 8 polymers-12-02758-f008:**
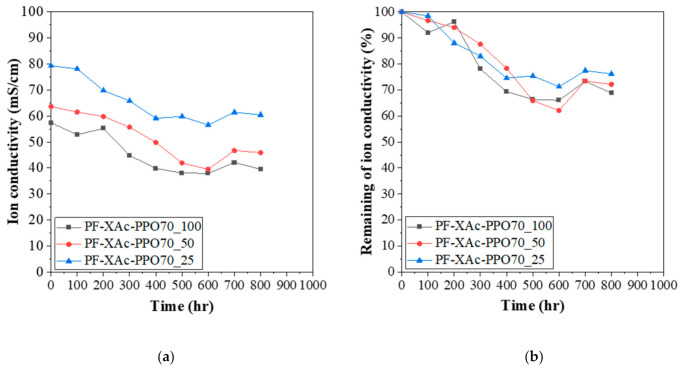
(**a**) The hydroxide conductivity (80 °C) and (**b**) the remaining of conductivity of crosslinked pore-filling membrane during immersion in aqueous 1M KOH at 60 °C.

**Figure 9 polymers-12-02758-f009:**
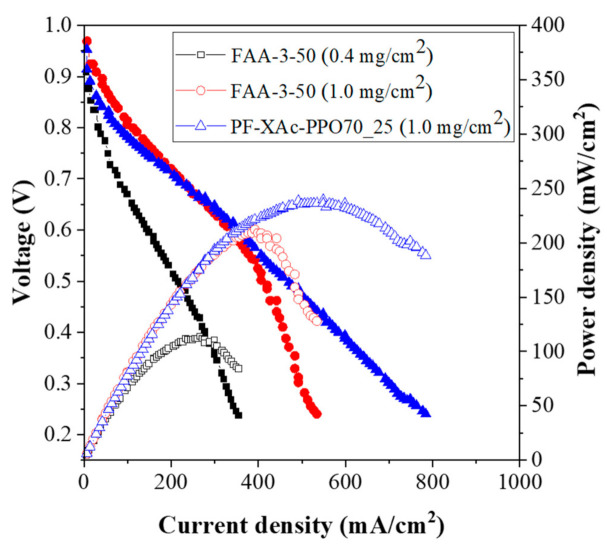
Cell test results of the XAc-PPO70_25 under H_2_/O_2_ conditions and 100% RH at 60 °C.

**Table 1 polymers-12-02758-t001:** Mechanical properties of crosslinked pore-filling membrane.

Properties	Sample			
	XAc-PPO70_100	PF-XAc-PPO70_100	PF-XAc-PPO70_50	PF-XAc-PPO70_25
Tensilte strength (MPa)	N/A	42.2	51.1	56.1
Elongation at break (%)	N/A	13.0	23.0	32.7
Young’s modulus (MPa)	N/A	1345.5	577.7	781.0

**Table 2 polymers-12-02758-t002:** The properties of crosslinked pore-filling membrane.

Sample	IEC (meq/g)		Water Uptake (%)		Swelling Ratio (%)				λ	
	Theo	Exp	25 °C	80 °C	25 °C (Δt)	80 °C (Δt)	25 °C (Δl)	80 °C (Δl)	25 °C	80 °C
XAc-PPO70_100	2.96	2.65 ± 0.12	106.0 ± 7.5	131.4 ± 5.5	7.38 ± 2.6	9.76 ± 0.2	29.2 ± 4.2	29.2 ± 4.2	19.9	24.7
PF-XAc-PPO70_100	N/A	2.05 ± 0.02	21.3 ± 1.9	57.7 ± 7.0	0.9 ± 0.9	0.9 ± 0.9	6.3 ± 1.3	9.0 ± 1.0	5.8	15.6
PF-XAc-PPO70_50	N/A	2.09 ± 0.03	37.0 ± 1.4	61.8 ± 8.1	2.7 ± 0.9	2.8 ± 0.9	9.3 ± 0.8	14.8 ± 0.8	9.8	16.4
PF-XAc-PPO70_25	N/A	2.05 ± 0.06	46.7 ± 1.3	78.9 ± 12.3	3.5 ± 1.8	6.7 ± 1.0	11.3 ± 1.3	17.5 ± 2.5	12.7	21.4

**Table 3 polymers-12-02758-t003:** Hydroxide conductivity of crosslinked pore-filling membranes.

Sample	Hydroxide Conductivity (mS/cm)					ASR (Ω·cm^2^)
	In DI water				95%RH	
	25 °C	40 °C	60 °C	80 °C	80 °C	
**PF-XAc-PPO70_100**	30.7	37.3	44.8	50.8	49.5	0.069
PF-XAc-PPO70_50	35.8	40.2	52.5	65.5	63.6	0.053
PF-XAc-PPO70_25	47.7	57.5	71.4	87.1	79.3	0.040

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
