# Peer review of "Crosslinked Pore-Filling Anion Exchange Membrane Using the Cylindrical Centrifugal Force for Anion Exchange Membrane Fuel Cell System"

_polymers, 2020, doi:10.3390/polym12112758_

Round 1
Reviewer 1 Report
The authors have characterized a class of crosslinked poly (phenylene oxide)-based membranes. Apparently, it has become a practical approach for researchers to improve the performance of the AEM by crosslinking method. This study follows the same trend to study their AEMs. However, the following points need to be addressed before the acceptance of the paper for publication in Polymers.
- In abstract, the author needs to summarize why XAc-PPO70_25 outperforms other candidates.
- In introduction, the author needs to elaborate why pore-filled membranes have better chemical stability than none pore-filled membranes.
- Again, the author needs to explain why crosslink the membrane could enhance the chemical stability of anion exchange membranes. What is the mechanism to improve chemical stability?
- For Figure 2, there is an unassigned peak between 2.25 ppm to 2.5 ppm. The author needs to clarify all peaks shown in NMR spectrum.
- In Figure 6, the author needs to provide clear scale bar to both stand-alone membranes and SEM images of membranes. To demonstrate that the substrate is uniformly filled with ionic material, it is better to conduct element mapping using EDX-SEM.
- The author needs to provide better explanation about how to correlate hydration number to water management. Please first define what is water management in AEMFC and how the change of hydration number help to improve the water management in AEMFC.
- The author needs to provide error bar for Figure 7.
- It is unclear for the audience why Xac-PPO70-25 has better alkaline stability than other candidates.
- In Figure 8, the author needs to explain why there is an increase in ion conductivity after 600 hours of immersion.
- The author should discuss fuel cell performance with leading AEMs in literature and make comments based on them.
Author Response
1) In abstract, the author needs to summarize why XAc-PPO70_25 outperforms other candidates.
: We thank to the reviewer for pointing out. In the abstract, the reason why PF-XAc-PPO70_25 electrolyte membrane is an excellent candidate has been modified and supplemented.
2) In introduction, the author needs to elaborate why pore-filled membranes have better chemical stability than none pore-filled membranes.
: In the case of recent ion exchange membranes, the chemical stability is improved by using phase separation, and since the use of a porous support can induce artificial phase separation, pore-filled membranes have been studied. We explained this by adding it to the introduction. (Page 2, line 58)
3) Again, the author needs to explain why crosslink the membrane could enhance the chemical stability of anion exchange membranes. What is the mechanism to improve chemical stability?
: Thank you for comments we can improve on a better paper. It has been supplemented by adding and modifying the contents below.
The crosslinking structure of PPO is responsible for the improvement in alkaline stability. As observed in Figure 8, PF-XAc-PPO70_25 exhibited about 76 % of initial conductivity value after exposed in 1M KOH at 60 °C for 800 h could be due to the steric hindrance offered by the crosslinking structure which can reduce the access of nucleophilic attack to the functional groups by the excess amount of OH− and protect the backbone (Crosslinked poly (2,6-dimethyl-1,4-phenylene oxide) polyelectrolyte enhanced with poly (styrene-b-(ethylene-co-butylene)-b-styrene) for anion exchange membrane applications - Zhihua Wang, Ziming Li, Nanjun Chen, Chuanrui Lu, Fanghui Wang, Hong Zhu, Journal of Membrane Science 564 (2018) 492–500). Moreover, the presence of water also have important role in hydroxide ion attack on membranes. PF-XAc-PPO70_25 is found to have a better alkaline stability than PF-XAc-PPO70_50 and PF-XAc-PPO70_100. The higher water uptake and improved solvation environment for OH- in PF-XAc-PPO70_25 would retard the degradation of cation and result in higher alkaline stability (Enhanced performance of anion exchange membranes via crosslinking of ion cluster regions for fuel cells - Ao Nan Lai, Dong Guo, Chen Xiao Lin, Qiu Gen Zhang, Ai Mei Zhu, Mei Ling Ye, Qing Lin Liu, Journal of Power Sources 327 (2016) 56-66).
4) For Figure 2, there is an unassigned peak between 2.25 ppm to 2.5 ppm. The author needs to clarify all peaks shown in NMR spectrum.
: We apologize for our mistakes. Peaks that were not assigned to 1H-NMR were peaks of unreacted material that appeared in the process of washing, and after washing several more times, it was confirmed that they disappeared in the 1H-NMR spectra measured again.
5) In Figure 6, the author needs to provide clear scale bar to both stand-alone membranes and SEM images of membranes. To demonstrate that the substrate is uniformly filled with ionic material, it is better to conduct element mapping using EDX-SEM.
: Thank you for comments. A photograph with a scale bar marked on the FE-SEM image was used, and the entire pore-filling was backed up through EDS image. The EDS image was explained at the supporting information.
6) The author needs to provide better explanation about how to correlate hydration number to water management. Please first define what is water management in AEMFC and how the change of hydration number help to improve the water management in AEMFC.
: The importance of water management in AEMFC and the relationship between hydration number and water management have been added. (Page 10, line 290 ~ 298)
7) The author needs to provide error bar for Figure 7.
: We thank to the reviewer for pointing out. We have added an error bar to Figure 7.
8) It is unclear for the audience why XAc-PPO70-25 has better alkaline stability than other candidates.
: We already discussed alkaline stability of PF-XAc-PPO70_25 in the page 11. PF-XAc-PPO70_25 has high ionic conductivity because it is easy to move hydroxide ions due to excellent water management due to long chains and chain flexibility due to relatively low crosslinking degree. For the same reason, in Figure 8 (b), it can be seen that the ionic conductivity of PF-XAc-PPO70_25 was maintained the most even after 800 hours in a strong alkaline atmosphere compared to other candidates.
9) In Figure 8, the author needs to explain why there is an increase in ion conductivity after 600 hours of immersion.
: In the case of long-term conductivity, not 1 sample (1 x 4 cm) was measured for 1000 hours, but 10 samples prepared under the same conditions were placed in 10 vials and measured every 100 hours. Therefore, it is a result that may appear during measurement, and this suggests a result that can be seen that the overall decline is occurring.
10) The author should discuss fuel cell performance with leading AEMs in literature and make comments based on them.
: Currently, a lot of research is being conducted due to the absence of leading AEM in the anion exchange membrane market. Therefore, we obtained a sample from Fumatech, which is currently on sale, and compared the cell performance in this work.
Reviewer 2 Report
The authors present a crosslinked pore-filling anion exchange membrane for AEMFC. This is an interesting piece of work. Hence, I find this suitable for publication in the journal after minor revision. The reviewer has the following suggestions for the authors to consider:
- Page 1, lines 42-43
The authors should reword the sentences and present them in a more positive way. There are a few articles (J. Mater. Chem. A, 2020, 8, 17568-17578; Adv. Energy Mater., 2020, 2001986; J. Electrochem. Soc., 2020, 167, 054501; J. Electrochem. Soc., 2019, 166, F637-F644; ACS Appl. Energy Mater., 2019, 2, 2458-2468; J. Membr. Sci., 2019, 570–571, 394-402) available which shows very high ionic conductivity, mechanical stability, alkaline stability, record-high fuel cell performances, and record-high cell durability. The authors should cite the articles mentioned above.
- Page 2, lines 47-50
Polynorbornene is the most important class of examples of AEMs considering their record high AEMFCs performances and durability. The authors should cite the following papers: J. Mater. Chem. A, 2020, 8, 17568-17578; Adv. Energy Mater., 2020, 2001986; J. Electrochem. Soc., 2020, 167, 054501; J. Electrochem. Soc., 2019, 166, F637-F644; ACS Appl. Energy Mater., 2019, 2, 2458-2468; J. Membr. Sci., 2019, 570–571, 394-402; ACS Appl. Energy Mater., 2020, 3, 4449-4456; ACS Appl. Energy Mater., 2019, 2, 2447-2457.
- Page 2, line 61
The authors should write SN2 as “SN2”.
- Figure 6 appeared in the manuscript before figure 4 and figure 5 in the manuscript. The authors should reorganize the text and the appearance of the figures in the manuscript.
- The authors should comment on the effect of thicknesses of membranes on water transport across the membranes (J. Electrochem. Soc. 2020, 167, 054501) during AEMFC operation.
- How about the in-situ cell durability of the AEMFC?
Author Response
- Page 1, lines 42-43
The authors should reword the sentences and present them in a more positive way. There are a few articles (J. Mater. Chem. A, 2020, 8, 17568-17578; Adv. Energy Mater., 2020, 2001986; J. Electrochem. Soc., 2020, 167, 054501; J. Electrochem. Soc., 2019, 166, F637-F644; ACS Appl. Energy Mater., 2019, 2, 2458-2468; J. Membr. Sci., 2019, 570–571, 394-402) available which shows very high ionic conductivity, mechanical stability, alkaline stability, record-high fuel cell performances, and record-high cell durability. The authors should cite the articles mentioned above.
: Thank you for the valuable comment. We revised it based on the comments you pointed out, and citation all references. [Ref. 12, 13, 14, 15, 16, 17]
- Polynorbornene is the most important class of examples of AEMs considering their record high AEMFCs performances and durability. The authors should cite the following papers: J. Mater. Chem. A, 2020, 8, 17568-17578; Adv. Energy Mater., 2020, 2001986; J. Electrochem. Soc., 2020, 167, 054501; J. Electrochem. Soc., 2019, 166, F637-F644; ACS Appl. Energy Mater., 2019, 2, 2458-2468; J. Membr. Sci., 2019, 570–571, 394-402; ACS Appl. Energy Mater., 2020, 3, 4449-4456; ACS Appl. Energy Mater., 2019, 2, 2447-2457.
: Thank you for being able to supplement our paper with good intelligence. We cited the reference paper you mentioned in the comment. [Ref. 12, 13, 14, 15, 16, 17, 31, 32]
- Page 2, line 61.
: We apologize for the oversight and thank to the reviewer for pointing out. We revised it to what reviewer pointed out.
- Figure 6 appeared in the manuscript before figure 4 and figure 5 in the manuscript. The authors should reorganize the text and the appearance of the figures in the manuscript.
: Thank you for the valuable point. We have reorganized the ordering arrangements that could confuse readers.
- The authors should comment on the effect of thicknesses of membranes on water transport across the membranes (J. Electrochem. Soc. 2020, 167, 054501) during AEMFC operation.
: While rearranging the order based on Comment 4, the importance of the thickness of the electrolyte membrane was added. The reference paper was also added.
- How about the in-situ cell durability of the AEMFC?.
: Thank you for your kindly comment. First of all, the purpose of this paper was not to look at cell performance because it focused on the preparation of crosslinked pore-filling membranes that are easily and uniformly using centrifugal force machine. I hope you will see the research paper on how to easily and uniformly adjust the thickness of crosslinked pore-filling membranes for polymer electrolyte membranes for fuel cells. Therefore, we're looking forward to expect that there will be a good opportunity to conduct fuel cell durability experiments with the studied crosslinked pore-filling membrane.
Reviewer 3 Report
This manuscript reported the preparation of crosslinked pore-filling AEMs using cylindrical centrifugal force. It is an interesting work. The obtained pore-filling AEMs showed high conductivity, excellent mechanical properties, as well as comparable alkaline stability. Moreover, the crosslinked pore-filling AEMs were used in AEM fuel cell achieving a peak power density of 239 mW/cm2. However, before I recommended its publication in Polymers, the following issues should be addressed.
(1) Introduction: the advantages of centrifugal force should be mentioned when it was used to fabricated pore-filled membranes.
(2) why Ac-PPO was selected to synthesize the anion conductive polymer? The carbonyl group between PPO and QA cations is not stable under alkaline condition, which have been reported previously.
(3) is there any change in thickness when PPO was filled in the PE membrane? how thick is the PE membrane?
(4) In Figure 2, the unassigned peaks should be clearly interpreted.
(5) Mechanical properties: it is not reasonable to use QAc-PPO70 membrane as a control sample to highlight the excellent strength of pore-filled AEMs, as QAc-PPO70 is not crosslinked.
(6) more discussion should be added in the degradation mechanism of the AEMs during alkaline stability.
Author Response
1) Introduction: the advantages of centrifugal force should be mentioned when it was used to fabricated pore-filled membranes.
: Thank you for the valuable point. We added to the introduction what advantages of using centrifugal force in preparing pore-filled membranes.
- why Ac-PPO was selected to synthesize the anion conductive polymer? The carbonyl group between PPO and QA cations is not stable under alkaline condition, which have been reported previously.
: After the acylation reaction, the reason why the carbonyl group was left unreduced is "rat trap strategy". In the paper reported previously, the enhanced stability can be attributed to the carbonyl group which can provide an alternative attacking path for hydroxide ion, which reduces the probability of the degradation of quaternary ammonium group. [J. Mater. Chem. A, 2017, 5, 6318-6327]
- Is there any change in thickness when PPO was filled in the PE membrane? how thick is the PE membrane?
: Crosslinked pore-filling membranes having a thickness of 30 um were prepared using a 20 um thick PE support. It was found that the prepared crosslinked pore-filling membranes were manufactured without significant thickness change. (Page 7)
- In Figure 2, the unassigned peaks should be clearly interpreted.
: We apologize for our mistakes. Peaks that were not assigned to 1H-NMR were peaks of unreacted material that appeared in the process of washing, and after washing several more times, it was confirmed that they disappeared in the 1H-NMR spectra measured again.
- Mechanical properties: it is not reasonable to use QAc-PPO70 membrane as a control sample to highlight the excellent strength of pore-filled AEMs, as QAc-PPO70 is not crosslinked.
: It was named QAc-PPO70 to distinguish it from the crosslinked pore-filling membrane, but it was actually a film produced by crosslinking. The sample has been re-named and clarified so that the reader can know exactly. (eg. QAc-PPO70 → XAc-PPO70_100, XAc-PPO70_100, 50, 25 → PF-XAc-PPO70_100, 50, 25)
- more discussion should be added in the degradation mechanism of the AEMs during alkaline stability.
: Thank you for comments we can improve on a better paper. We added and supplemented the information you pointed out.
Round 2
Reviewer 1 Report
The paper can be published in its current form.
Reviewer 3 Report
The manuscript could be accepted for publishing after the revision.